# Comprehensive Taxonomical Analysis of *Trichophyton mentagrophytes/interdigitale* Complex of Human and Animal Origin from India

**DOI:** 10.3390/jof9050577

**Published:** 2023-05-16

**Authors:** Shivaprakash M. Rudramurthy, Dipika Shaw, Shamanth Adekhandi Shankarnarayan, Sunil Dogra

**Affiliations:** 1Department of Medical Microbiology, Postgraduate Institute of Medical Education and Research, Chandigarh 160012, India; 2Department of Microbiology, Indian Veterinary Research Institute, Izatnagar, Bareilly 243122, India; 3Department of Dermatology, Venerology and Leprology, Postgraduate Institute of Medical Education and Research, Chandigarh 160012, India

**Keywords:** *Trichophyton mentagrophytes*, phylogenetic analysis, internal transcribed spacer, translational elongation factors, mating gene, animal

## Abstract

Taxonomic delineation of etiologic agents responsible for recalcitrant dermatophytosis causing an epidemic in India is still debated. The organism responsible for this epidemic is designated as *T. indotineae*, a clonal offshoot of *T. mentagrophytes*. To evaluate the real identity of the agent causing this epidemic, we performed a multigene sequence analysis of *Trichophyton* species isolated from human and animal origin. We included *Trichophyton* species isolated from 213 human and six animal hosts. Internal transcribed spacer (ITS) (*n* = 219), translational elongation factors (*TEF 1-α*) (*n* = 40), ß-tubulin (BT) (*n* = 40), large ribosomal subunit (LSU) (*n* = 34), calmodulin (CAL) (*n* = 29), high mobility group (*HMG*) transcription factor gene (*n* = 17) and α-box gene (*n* = 17) were sequenced. Our sequences were compared with *Trichophyton mentagrophytes* species complex sequences in the NCBI database. Except for one isolate (ITS genotype III) from animal origin, all the tested genes grouped our isolates and belonged to the “Indian ITS genotype”, currently labeled as *T. indotineae*. ITS and *TEF 1-α* were more congruent compared to other genes. In this study, for the first time, we isolated the *T mentagrophytes* ITS Type VIII from animal origin, suggesting the role of zoonotic transmission in the ongoing epidemic. Isolation of *T. mentagrophytes* type III only from animal indicates its niche among animals. Outdated/inaccurate naming for these dermatophytes in the public database has created confusion in using appropriate species designation.

## 1. Introduction

Dermatophytosis due to *Trichophyton* species has reached an epidemic stage in the Indian subcontinent [1]. Probable reason for such a massive epidemic in India remains enigmatic. Taxonomic delineation of etiologic agents may provide leading evidence in understanding this ongoing epidemic. Labeling the etiological agent as *T. mentagrophytes* or *T. interdigitale*, or *T. indotineae* is still debated.

Two studies from India in 2018 attributed *T. interdigitale* (identified based on the classification of *Trichophyton* species described by de Hoog et al.) as the predominant species causing dermatophytosis [2,3,4,5]. In both studies, identity was confirmed as *T. interdigitale* (an anthropophilic species) based on the internal transcribed spacer (ITS) region of rDNA [4,5]. Nenoff et al. analyzed the ITS sequences of *T. interdigitale* and iterated that these isolates were *T. mentagrophytes* ITS genotype VIII (Indian genotype) and are zoophilic [6]. Chowdhary et al. indicated that rearing a pet in India is a rare phenomenon, and the zoophilic spread is questionable. Further, based on the concatenated ITS and β-tubulin sequences, they emphasized that the etiology was anthropophilic fungus [7]. Later, Nenoff et al. hypothesized that either zoophilic or geophilic *Trichophyton* species underwent “anthropization”, developing high virulence under a suitable skin milieu. Based on in silico analysis, Pchelin et al. indicated that the Indian isolate included in their study was sister species to *T. mentagrophytes* and *T. interdigitale* [8]. Similarly, based on whole genome sequencing of the epidemic isolates, Singh et al. concluded that it was difficult to differentiate this epidemic strain from *T. mentagrophytes/T. interdigitale* complex [9]. Recently, Kano et al. isolated dermatophytes from patients of Indian origin in Japan and renamed them *T. indotineae* [10]. These isolates corresponded to Indian genotype VIII, exhibited high terbinafine minimum inhibitory concentration (MIC) and possessed unique clinical and mycological features [10]. Later, Tang et al. concluded that Indian genotype VIII differs from *T*. *mentagrophytes sensu stricto* and *T. interdigitale sensu stricto* [11]. They also highlighted that as the phenotypic and physiologic characteristics do not significantly vary from the name change has practical rather scientific justification [11].

The accurate identification of the Indian epidemic strain causing dermatophytosis is still uncertain. Hence, in the present study, we performed a multigene sequence of *Trichophyton* species isolated from human and animal origin and compared it with the published database to highlight the discrepancies in the labeling of the *T. mentagrophytes/T. interdigitale* complex in the GenBank database.

## 2. Material and Methods

### 2.1. Fungal Strain and Growth Condition

A total of 213 *Trichophyton mentagrophytes/T. interdigitale* complex species isolated from dermatophytosis cases at our tertiary care referral hospital in India were used in this study. Six *T. mentagrophytes/T. interdigitale* complex species of animal origin (canine) included in the study were isolated at Indian Veterinary Research Institute (IVRI), Izatnagar, India. The isolates were revived on Sabouraud Dextrose Agar (SDA) (HiMedia, Mumbai, India) containing chloramphenicol (0.05 mg/L) and cycloheximide (0.5 mg/L) and incubated at 28 °C.

### 2.2. DNA Extraction, Amplification and Sequencing

The genomic DNA was extracted by the phenol–chloroform–isoamyl alcohol method as previously standardized in our laboratory [4]. Polymerase Chain Reaction (PCR) was performed for Internal transcribed spacer (ITS) (*n* = 219), translational elongation factors *(TEF 1-α*) (*n* = 40), ß-tubulin (BT) (*n* = 40), large ribosomal subunit (LSU) (*n* = 34), calmodulin (CAL) (*n* = 29), high mobility group (*HMG*) transcription factor gene (*n* = 17) and α-box gene (*n* = 17). The primer pairs used for amplifying seven genes are provided in Appendix A. We amplified the genes in the presence of 1X Taq polymerase buffer with 2 mM MgCl_2_, 200 mM each dNTP (Genei Laboratories Pvt. Ltd., Bengaluru, India), 0.2 mM each primer (Integrated DNA Technologies), 0.3 U Taq polymerase (Genei Laboratories Pvt. Ltd., Bengaluru, India) and 50–100 ng fungal genomic DNA. PCR cycling conditions consisted of an initial denaturation step for 5 min at 95 °C followed by 35 cycles of 94 °C for 1 min, 58 °C (Tm for *TEF 1α*, BT, LSU and CAL)/56 °C (Tm for ITS, *HMG* and α-box) for 30 s and 72 °C for 1 min, with a final extension step at 72 °C for 7 min. Sequencing PCR was performed for both forward and reversed strands with the primers mentioned above and BigDye Terminator Cycle sequencing kit version 3.1 (Applied Biosystems, Foster City, CA, USA). All the sequencing reaction products were purified and analyzed on an ABI 3500 genetic analyzer (Applied Biosystems). The consensus sequence was generated using Seqman software (DNASTAR, Lasergene). Sequences were compared with the GenBank DNA database using the BLAST tool, the ISHAM ITS database and the CBS database. [https://blast.ncbi.nlm.nih.gov (accessed on 21 September 2021), http://its.mycologylab.org/BioloMICSSequences.aspx (accessed on 21 September 2021) and http://www.westerdijkinstitute.nl/Collections/BioloMICSSequences.aspx (accessed on 21 September 2021)].

### 2.3. Phylogenetic Analysis

All the sequences of ITS, *TEF 1-α*, BT, LSU, CAL, *HMG* and α-box used previously in the literature from 2017 for identifying *Trichophyton mentagrophytes*/*T. interdigitale* complex was retrieved from the NCBI database [3,6,7,8,11,12,13,14]. *T. benhamiae* (CBS 623.66) was used as the outgroup. In the present study, for barcoding, the ITS region, whole ITS1, 5.8 regions and ITS2 of KT354634 GenBank accession having 593 base pairs was used [12,15]. Retrieved sequences from the NCBI database and our sequences were aligned using multiple sequence alignment modes in ClustalX2 software. We exported the aligned sequences to Molecular Evolutionary Genetics Analysis software version X (MEGA X) [16]. The neighbor-joining tree was constructed with 1000 bootstrapping replicates using the Kimura 2 parameter model. Modification of the phylogenetic tree graphical representations was done by the Fig Tree (version 1.4.4) and Dendroscope 3 software (version 3.7.5) [17,18].

### 2.4. Highlighting the Discrepancies in the Database

From 2017 to date, the sequences of the isolates used to delineate the taxonomy of *T. mentagrophytes/T. interdigitale* complex (*n* = 457) was retrieved to verify the identity submitted in the GenBank database. The retrieved sequences of ITS gene (*n* = 374), *TEF1-α* (*n* = 184), BT (*n* = 52), LSU (*n* = 50), CAL (*n* = 69) and *HMG* transcription factor (*n* = 102) were validated as per the phylogenetic analysis of ITS gene in the present study.

## 3. Results

Phylogenetic analysis of the ITS gene revealed that, except for one isolate from animal origin, all other isolates [human (*n* = 213) and animal origin (*n* = 5)] belonged to *T. mentagrophytes* ITS genotype VIII. The remaining isolate from animal origin belonged to ITS genotype III. The phylogenetic tree constructed based on the Neighbor-Joining method using Kimura 2- parameter model had a total of 15 clusters for *T. mentagrophytes* with previously defined genotypes III, III*, IV, V, VII, VIII, IX, XI, XII, XIII, XV-XVI, XVIII, XXII and an undefined genotype (KMU 5471, AB617775). Whereas *T. interdigitale* consisted of a cluster previously defined as Type I, II and II* and *T. interdigitale* genotype X (Table 1) (Appendix A).

Like the ITS gene, the *TEF 1-α* gene also clustered all our isolates [human (*n* = 34) and animal (*n* = 5) origin] together along with the *T. mentagrophytes* ITS genotype VIII isolates. In contrast, the remaining isolate from animal origin (PGI-IVRI B24-A) clustered with *T. mentagrophytes* ITS genotype III, III*, V and *T. interdigitale*. Phylogenetic analysis based on *TEF 1-α* divided isolates into six clusters (cluster 1: *T. mentagrophytes* ITS genotype VIII; cluster 2: *T. mentagrophytes* ITS genotype III, III*, V and *T. interdigitale*; cluster 3: *T. mentagrophytes* ITS genotype IV; cluster 4: *T. mentagrophytes* ITS genotype IV; cluster 5: *T. mentagrophytes* ITS genotype III, VIII and *T. interdigitale*; cluster 6: *T. mentagrophytes* genotype III*, VII and IX) (Appendix A).

The phylogenetic tree based on the BT gene (*n* = 39) also clustered all our isolates and isolates belonging to *T. mentagrophytes* ITS genotype IV, VIII and *T. interdigitale*. The remaining isolate from animal origin (PGI-IVRI B24-A) formed a separate cluster and did not merge with other genotypes. Based on the BT gene, all the analyzed isolates were grouped into four clusters (cluster 1: *T. mentagrophytes* type IV, VIII and *T. interdigitale*; cluster 2: PGI-IVRI B24-A; cluster 3: *T. mentagrophytes* type III, III*, V, VII; and cluster 4: *T. mentagrophytes* type III*, IV and *T. interdigitale* (Appendix A).

Similar to other genes, LSU also clustered all our isolates (*n* = 34) together. Isolate PGI-IVRI B24-A grouped with *T. mentagrophytes* ITS genotype III and III* isolates. Based on the LSU gene, all the analyzed isolates grouped into seven clusters (cluster 1: *T. mentagrophytes* type III and III*; cluster 2: *T. mentagrophytes* type IV, V and *T. interdigitale*; cluster 3: *T. mentagrophytes* type VII; cluster 4: *T. interdigitale*; cluster 5: *T. mentagrophytes* type IV; cluster 6: *T. mentagrophytes* type VIII; and cluster 7: *T. interdigitale*) (Appendix A).

CAL gene clustered all our isolates (*n* = 28) together and grouped two *T. interdigitale* (KM387164/KM387182) isolates. The animal isolate (PGI-IVRI B24-A) grouped with *T. mentagrophytes* type VII (Appendix A).

Except for one animal isolate (PGI-IVRI B24-A), all the isolates of *Trichophyton* species from human (*n* = 11) and animal origin (*n* = 5) contained *HMG* transcription factor but lacked α-box genes. The animal isolate (PGI-IVRI B24-A) had both *HMG* transcription factor and α-box genes suggesting its homothallic nature. Phylogenetic analysis based on the *HMG* transcription factor grouped 16 isolates (11 from humans and five from animals) in cluster-1, whereas animal isolate PGI-IVRI B24-A was grouped in cluster-2. All the isolates analyzed were grouped into five clusters (cluster 1: *T. mentagrophytes* type VIII; cluster 2: *T. mentagrophytes* type III, III*, IV, IX; cluster 3: *T. interdigitale*; cluster 4: *T. mentagrophytes* type III, III* and IX and *T. interdigitale*; and cluster 5: *T. interdigitale*) (Appendix A).

A comprehensive analysis of the labeling of submitted sequences in the GenBank database revealed discrepancies. The accurate identification of the isolates was considered based on our phylogenetic analysis (ITS gene). Based on the analysis, a large number of 4 ITS sequences deposited in the GenBank databases were mislabeled, i.e., *T. mentagrophytes* was named as *T. interdigitale* or vice versa (Appendix A).

## 4. Discussion

The identity of the pathogen responsible for the dermatophytosis epidemic in India is under scrutiny. The ongoing outbreak of superficial dermatophytosis is mainly due to *Trichophyton* spp. The *Trichophyton mentagrophytes/T. interdigitale* complex is the principal etiological agent responsible, and resistance in these isolates commonly occurs and leads to treatment failure. Thus finding the etiological agent may provide better insight into therapeutic management [9]. The whole-genome sequence of Indian *Trichophyton* spp. (D15P135) formed a conspecific clade distant from *T. mentagrophytes* and *T. interdigitale* [8]. Based on the ITS phylogenetic tree, Nenoff et al. designated the Indian isolates as ITS genotype VIII [6]. Several other studies have also reported that *T. mentagrophytes* ITS type VIII is most common in India [9,13,14,20] and second most common in Iran [14]. Apart from India and Iran, other countries (i.e., Japan, Germany, Oman and France) also reported *T. mentagrophytes* ITS genotype VIII [10,11,19].

Further, based on whole-genome analysis, Singh et al. showed that all the clinical isolates of India belonged to *T. mentagrophytes* type VIII and had only 42 SNPs between any two isolates. They could not conclusively differentiate *T. mentagrophytes* from *T. interdigitale* [9]. Recently, Kano et al. described two isolates of Indian origin resistant to terbinafine as a new species, *T. indotineae*, based on the ITS gene sequence. However, later Tang et al. performed multigene phylogeny (ITS, *TEF 1-α* and *HMG* gene) of many isolates *(n =* 182) and verified that it was indeed a new species. In the present study, we found that all the clinical (human) isolates formed a cluster with *T. mentagrophytes* ITS genotype VIII (D15P135; accession no KY761968).

Nenoff et al., in 2019, introduced a total of nine genotypes, including *T. interdigitale* (genotype I and II) and *T. mentagrophytes* (genotype III to IX) based on ITS sequences [19] and indicated that Indian origin isolates were *T. mentagrophytes* ITS genotype VIII. Further, Taghipour et al. analyzed their sequences including the sequences of IX genotypes (described by Nenoff et al.) and described more genotypes (genotype X to XXIV) [14]. Recently, Nenoff et al. also introduced one more genotype (genotype XXV) [13]. In the present study, similar to the study by de Hoog et al., the ITS gene differentiated type and neotype strain of *T. mentagrophytes* (CBS 646.73^T^, CBS 304.38^T^, CBS 126.34^T^ and IHEM 4268^NT^) from *T. interdigitale* (CBS 647.73^T^, CBS 425.63^T^ and CBS 428.63^NT^). We also found that the Type and neotype strain of *T. mentagrophytes* belongs to ITS genotype III* (IHEM 4268^NT^ and CBS 126.34^T^) and ITS genotype IV (CBS 646.73^T^ and CBS 304.38^T^), respectively. Whereas all the type and neotype strains (CBS 647.73^T^, CBS 425.63^T^ and CBS 428.63^NT^) of *T. interdigitale* belong to a single cluster of *T. interdigitale.*

The phylogenetic analysis of BT sequences revealed that Indian isolate were similar to the Type strains of *T. interdigitale* defined by de Hoog et al.; CBS 425.63^T^ (MF898380), and CBS 647.73^T^ (KT155595). [3]. However, Type isolates defined as *T. mentagrophytes* CBS 304.38^T^ (MF898372) by de Hoog et al. [3] and JF731044 (*T. mentagrophytes* ITS genotype VIII) by Nenoff et al. [6] were also clustered with our isolates [3]. Therefore, the BT sequences appear to have less discriminatory power than ITS sequences, leading to Chowdhary et al. concluding all the isolates as *T. interdigitale.* [7] LSU gene could not even differentiate the type strain of *T. mentagrophytes* KT155300 (CBS 646.73^T^) from *T. interdigitale* (type and neotype strains), thereby questioning their validity in identifying these species correctly. Like Tang et al. and Nenoff et al., ITS and TEF 1alpha were more congruent, with more diversity in the ITS gene [11,13].

In the present study, we found many previously designated isolates with one genotype belonging to other genotype clusters (Appendix A). For example, the sequence of isolates described as genotype VI (DP15P161, D15P156) clustered with genotype IV isolates in our study [12,13]. Similarly, the new genotype introduced by Taghipour et al., i.e., genotype XVII (RezRaf-277, MK312990), clustered with genotype III* [12]. The other genotypes by Taghipour et al., XX (S-542, MK313030), XXI (36_S, MK312891), XXIII (ZM1, MK313044) clustered with genotype V; XXIV (RV 27961, AF170453) grouped with genotype IV; *T. mentagrophytes* genotype XXIV (JCM 1891, JN134094) belong to genotype type IV (Appendix A) [12]. The new genotype XXV introduced by Nenoff et al. (218292/17, MN886815; 201341/18, MN886816) clustered with genotype III* [13]. Further, as per Nenoff et al., *T. interdigitale* II* and X isolates form a separate group from genotypes I and II. Still, in the present study, these isolates belong to a single group in which all the *T. interdigitale* genotypes I and II are present (Appendix A) [13]. The confusion in identification is due to a need for more upgradation in strain details deposited in the NCBI and CBS databases. Few of the *Trichophyton* spp. sequence details are incorrect (Appendix A), which has also been emphasized by Chowdhary et al. [7]

Two isolates (IHEM 10162 and JCM 1891), which were previously described as *T. mentagrophytes* genotype IV based on ITS, form a separate cluster (cluster 3-Appendix A) as per the *TEF 1-α* gene analysis in the present study [12]. Further, due to the unavailability of gene bank accession numbers for the type and neotype strain of *T. mentagrophytes* and *T. interdigitale*, we could not assess whether this gene could differentiate them. Only the neotype strain of *T. mentagrophytes* (IHEM 4268^NT^) sequence was available (ITS genotype III* based on ITS gene), which formed a cluster with isolates of *T. mentagrophytes* ITS genotype III, III*, V, and *T. interdigitale* (CBS 130806) strain (Appendix A).

Recently, Tang et al. suggested that the presence of *HMG* transcription factor without α-box may provide the answer to identify the etiological agent responsible for the ongoing epidemic due to *T. indotineae*. *HMG* and α box are the drivers of evolution among dermatophytes [21]. *T. mentagrophytes* is the ancestral species from which ‘clonal offshoots’ *T. interdigitale* and *T. indotineae* were derived [11]. The occurrence of α box indicates the presence of MAT1-1 mating type, whereas the presence of MAT1-2 mating is indicative of the *HMG* domain [22]. Previously, Persinoti et al. reported that all the *T. rubrum* and *T. interdigitale* isolates used in their study showed the presence of a single mating type and all the population are clonally similar with a low level of diversity [22]. Further, based on the *HMG* gene, they described *T. rubrum* and *T. interdigitale* (causing an ongoing epidemic in India or *T. indotineae*) as anthropophilic. Anthropophilic dermatophytes have lost their sexuality [23], as seen among *T. interdigitale*. Similar to Nenoff et al. and Tang et al., all our human isolates (*n* = 11) belong to the “plus mating type” and carry only the *HMG* transcription factor at the MAT locus suggesting that the ongoing epidemic in India was due to a single type of mating strain (i.e., *HMG* transcription factor). Further, this finding shows agreement with the phylogenetic data of ITS and TEF1-α in which all the ongoing Indian epidemic isolates cluster together [11,19].

Chowdhary et al. reinforced that all the isolates used in the study by Singh et al. were *T. interdigitale* based on concatenated phylogeny of ITS and beta-tubulin sequences [7]. Further, they also reported that the patient presented with an atypical clinical appearance suggesting more like zoophilic infection. Thus, the question is whether the disease in India is due to the zoophilic *T. mentagrophytes* or anthropophilic *T. interdigitale* species. In our experience, of the 195 patients included in our previous study, only 28 (14.3%) gave a history of possible animal contact [4]. This is in full agreement with the study by Khurana et al. in which among 64 (75.3%) culture positive cases, only 2 (3.12%) patients had a history of animal contact [24]. We also agree that rearing a pet in India is relatively rare [7].

We did not find any nucleotide variation in the ITS and TEF sequences of human and animal origin (five isolates) *T. mentagrophytes* type VIII isolates. The animal-origin isolates of *T. mentagrophytes* type VIII only had plus mating-type of *HMG* transcription factor and did not show the presence of α- box, suggesting its nature like human-origin isolates. Till date, all the reported *T. mentagrophytes* ITS genotype VIII in the literature isolated from a human source, but in the present study, for the first time, we isolated *T. mentagrophytes* ITS genotype VIII from animal origin (pet dogs). Furthermore, several isolates are required to support the validity of the circulated anthropophilic *T. indotineae* among the animals.

A possible solution to resolve the confusion is to curate the sequences deposited in the public databases. There is also a need to define barcode sequence, positions (trimmed ends) of ITS to designate the *Trichophyton* species. An international consortium to discuss the taxonomy and designating neotype or type strain for any new species or subspecies (based on housekeeping gene sequence or multiple gene sequences) may help resolve the taxonomy issues. Further, the full genome sequencing project on dermatophytes might help to resolve the current taxonomy issues.

## Figures and Tables

**Table 1 jof-09-00577-t001:** Salient features of the phylogenetic analysis of representative sequences of different *Trichophyton* ITS genotype (retrieved from NCBI database) and isolates from the present study.

Sl No.	Strain Number	Country	Source	Genotype Given as Per the Literature	Accession Number ITS	Naming as Per Current Study	Reference
1	A148	Iran	H	Tm type XIII	MK312917	Tm type XIII	[14]
2	211497/17	India	H	Tindo/Tm type VIII	NA	Tm type VIII	[11]
3	211501/17_DSM 107596	India	H	Tindo/Tm type VIII	MH791419	Tm type VIII	[9,11,19]
4	211509/17	India	H	Tindo/Tm type VIII	MH791420	Tm type VIII	[11]
5	216500/17_DSM 107599	India	H	Tindo/Tm type VIII	MH791422	Tm type VIII	[11,19]
6	MYD 2	India	H		MN831065	T. indo/ITS genotype VIII Tm	This study
7	MYD 5	India	H		MN831089	T. indo/ITS genotype VIII Tm	This study
8	MYD 11	India	H		MN831064	T. indo/ITS genotype VIII Tm	This study
9	MYD 12	India	H		MN831088	T. indo/ITS genotype VIII Tm	This study
10	MYD 27	India	H		MH517546	T. indo/ITS genotype VIII Tm	This study
11	MYD 33	India	H		MN831103	T. indo/ITS genotype VIII Tm	This study
12	MYD 38	India	H		MN831039	T. indo/ITS genotype VIII Tm	This study
13	MYD 43	India	H		MN831087	T. indo/ITS genotype VIII Tm	This study
14	MYD 47	India	H		MN831038	T. indo/ITS genotype VIII Tm	This study
15	MYD 50	India	H		MH517547	T. indo/ITS genotype VIII Tm	This study
16	MYD 51	India	H		MN831037	T. indo/ITS genotype VIII Tm	This study
17	MYD 53	India	H		MN831086	T. indo/ITS genotype VIII Tm	This study
18	MYD 54	India	H		MH517557	T. indo/ITS genotype VIII Tm	This study
19	MYD 55	India	H		MN831102	T. indo/ITS genotype VIII Tm	This study
20	MYD 59	India	H		MN831040	T. indo/ITS genotype VIII Tm	This study
21	MYD 60	India	H		MH517548	T. indo/ITS genotype VIII Tm	This study
22	PGI_IVRI_NCCPF:800068	India	A			T. indo/ITS genotype VIII Tm	This study
23	PGI_IVRI_B9_20A	India	A			T. indo/ITS genotype VIII Tm	This study
24	PGI_IVRI_B9_6A	India	A			T. indo/ITS genotype VIII Tm	This study
25	PGI_IVRI_B11_3A	India	A			T. indo/ITS genotype VIII Tm	This study
26	PGI_IVRI_NCCPF:800067	India	A		MN108151	T. indo/ITS genotype VIII Tm	This study
27	IHEM 4268NT ™	Belgium	H	Tm type III/III*	MF926358/JQ407193	Tm type III*	[11,14]
28	V34-22	Netherlands	H	Tm type III*	MW346113	Tm type III*	[11]
29	217907/15_DSM 108628	Germany	H	Tm type III*	MK447605	Tm type III*	[6,11,19]
30	218893/16_DSM 108629	Germany	H	Tm type III*	MK447604	Tm type III*	[6,11,19]
31	MJN-120	Iran	H	Tm type XV	MK312937	Tm type XV and XVI	[14]
32	MJN-98	Iran	H	Tm type XVI	MK312933	Tm type XV and XVI	[14]
33	XM11	China	H	Tm type IX	MW346058	Tm type IX	[11]
34	XM19	China	H	Tm type IX	MW346065	Tm type IX	[11]
35	XM1	China	H	Tm type IX	MW346048	Tm type IX	[11]
36	XM3	China	H	Tm type IX	MW346050	Tm type IX	[11]
37	KMU 5471	Japan	H	Tm	AB617775	Tm*	[14]
38	200128/17_DSM 108623	Germany	H	Tm type VII	MK447611	Tm type VII	[11,19]
39	215003/16_DSM 108624	Germany	H	Tm type VII	MK450324	Tm type VII	[6,11,19]
40	218904/16_DSM 108622	Germany	H	Tm type VII	MK450322	Tm type VII	[6,11,19]
41	210363/16_DSM 108625	Germany	H	Tm type VII	MK450323	Tm type VII	[6,11,19]
42	CBS 642.73_ATCC 28146	Netherlands	NA	Tm type IV	KJ722759	Tm type IV	[11]
43	IHEM 22740	Switzerland	H	Tm type IV	GU929694	Tm type IV	[6,11,19]
44	V296-57	Italy	A	Tm type IV	MW346154	Tm type IV	[11]
45	204543/17_DSM 108626	Germany	H	Tm type IV	MK447609	Tm type IV	[6,11,19]
46	600024/20	Iraq	H	Tm type V	MT374269	Tm type V	[6]
47	600014/20	Iraq	H	Tm type V	MT374268	Tm type V	[6]
48	600316/19	Iran	H	Tm type V	MT374257	Tm type V	[6]
49	600197/19	Iran	H	Tm type V	MT374258	Tm type V	[6]
50	27_S	Iran	H	Tm type XXII	MK312888	Tm type XXII	[14]
51	S74	Iran	H	Tm type XVIII	MK313028	Tm type XVIII	[14]
52	PFCC93-417	NA	NA	Ti type XII	MF109039	Ti type XII	[14]
53	217704/17_DSM 108630	Switzerland	A	Tm type III	MK450325	Tm type III	[6,11,19]
54	RCPF 1207	Russia	H	Tm type III	KT253559	Tm type III	[6,19]
55	200002/16_DSM 103451	Switzerland	A	Tm type III	KX866689	Tm type III	[6,19]
56	ATCC 60612	NA	NA	Tm type III	KJ606099	Tm type III	[6,19]
57	PGI_IVRI_B9_24A	India	A			Tm type III	This study
58	Rez-437	Iran	H	Ti type XI	MK312755	Ti type XI	[14]
59	V296-56	India	A	Ti	MW346155	Ti	[11]
60	V296-59	India	A	Ti	MW346153	Ti	[11]
61	V296-58	India	A	Ti	MW346152	Ti	[11]
62	A238	Australia	H	Ti	MW346178	Ti	[11]

Tm—*Trichophyton mentagrophytes*, Ti—*Trichophyton interdigitale*, Tindo—*Trichophyton indotineae*, Tm type III*—*Trichophyton mentagrophyte*s type III genotype-, H—Human skin scrapings, A—Animal skin scraping (Dog), ITS—Internal transcribed spacer.

## Data Availability

All data were available in NCBI database accession number provided in manuscript (Url: https://www.ncbi.nlm.nih.gov/ accessed on 21 September 2021).

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
