# Peer review of "Comprehensive Taxonomical Analysis of Trichophyton mentagrophytes/interdigitale Complex of Human and Animal Origin from India"

_jof, 2023, doi:10.3390/jof9050577_

Round 1

Reviewer 1 Report (Previous Reviewer 1)

The revised manuscript is corresponded to their aims to indicate that Trichophyton mentagrophytes. typeVIII can be found in both human and animal in India.

One small issue is about the typing error which should be carefully checked.

Author Response

Comment: One small issue is about the typing error which should be carefully checked.

Reply: Thank you for highlighting this issue, we have rectified the typing errors in this revision.

Reviewer 2 Report (New Reviewer)

The main question addressed by the research is taxonomical analysis of Trichophyton mentagrophytes/interdigitale complex  The gap is difficult to carry out the Taxonomical analysis. The subject of the manuscript is very specific about taxonomy Trichophyton mentagrophytes/interdigitale complex.

The authors should consider to compare the PCR analysis with reference strains of Trichophyton mentagrophytes/interdigitale ATCC. The conclusions are consistent with the evidence and arguments presented. But the worst problem is the absence of dermatophytes complete  sequencing program.

Introduction: Please cite the manuscript  

Recent advances in the diagnosis of dermatophytosis.

Begum J, Mir NA, Lingaraju MC, Buyamayum B, Dev K.

J Basic Microbiol. 2020 Apr;60(4):293-303. doi: 10.1002/jobm.201900675. Epub 2020 Jan 31.

PMID: 32003043 Review.   77- The primer pairs used for amplifying seven genes are provided in Supplementary Table S1 (Please insert Table S1)   A possible solution to resolve the confusion is to curate the sequences deposited in 269 the public databases. There is also a need to define barcode sequence, positions (trimmed 270 ends) of ITS to designate the Trichophyton species. An international consortium to discuss 271 the taxonomy and designating neotype or type strain for any new species or subspecies 272 (based on housekeeping gene sequence or multiple gene sequences) may help resolve the 273 taxonomy issues. Please insert coment about the full sequencing project of dermatophytes 

Author Response

Comment 1: The authors should consider to compare the PCR analysis with reference strains of Trichophyton mentagrophytes/interdigitale ATCC. The conclusions are consistent with the evidence and arguments presented. But the worst problem is the absence of dermatophytes complete sequencing program.

Reply: For ITS analysis, we included the reference strain CBS 642.73_ATCC 28146 (T. mentagrophytes type IV) previously used by Tang et al. Like Tang et al., these isolates clustered with  T. mentagrophytes type IV group in our analysis. Further, as per the reviewer's suggestion, using Trichophyton mentagrophytes/interdigitale ATCC strain is better to differentiate this problem. But, there are no ATCC stains for Trichophyton mentagrophytes/interdigitale genotype VIII.

Comment 2: Introduction: Please cite the manuscript. Recent advances in the diagnosis of dermatophytosis. Begum J, Mir NA, Lingaraju MC, Buyamayum B, Dev K. J Basic Microbiol. 2020 Apr;60(4):293-303. doi: 10.1002/jobm.201900675. Epub 2020 Jan 31. PMID: 32003043 Review.  

Reply: As per the reviewer's suggestion, the reference has been inserted in the revised version.

Comment 3: The primer pairs used for amplifying seven genes are provided in Supplementary Table S1 (Please insert Table S1) 

Reply: Supplementary Table S1 is now provided in the supplementary file.

Comment 4: A possible solution to resolve the confusion is to curate the sequences deposited in public databases. There is also a need to define the barcode sequence and positions (trimmed ends) of ITS to designate the Trichophyton species. An international consortium to discuss the taxonomy and designating neotype or type strain for any new species or subspecies (based on housekeeping gene sequence or multiple gene sequences) may help resolve the taxonomy issues. Please insert a comment about the full sequencing project of dermatophytes.

Reply: We agree with the reviewer and added this point in the revised manuscript.

Reviewer 3 Report (New Reviewer)

A few spelling mistakes

Author Response

Comment 1: Can the authors state how and when they received the animal isolates – where are these clinical isolates from infected dogs?

Reply: The source of animal isolates is mentioned in the revised manuscript (material and method part).

Comment 2: HMG and ? box are the drivers of evolution among dermatophytes. This line needs a little more explanation for readers.

Reply: Explanations related to HMG and ? box were added to the manuscript.

Comment 3: Is it possible at this stage to estimate when the Indian ITS genotype is likely to have diverged

Reply: With the data we generated, it is difficult to comment on the divergence of the Indian ITS genotype.

Comment 4: A few spelling mistakes

Reply: Thank you for highlighting the issues; we have modified the spelling mistake.

This manuscript is a resubmission of an earlier submission. The following is a list of the peer review reports and author responses from that submission.

Round 1

Reviewer 1 Report

The authors investigated the identity of human and animal Trichophyton in India based on multigene sequences in the regions of ITS, TEF 1 alpha, BT, LSU, CAL, HMG transcription factor, and alpha-box primers. The evidence of the common anthropophilic Trichophyton-ITS type VIII or Indian ITS genotype, is first found as zoophilic type in India, implied its a niche among animals there.  

There is one missing for the legend of Figure 4 , LSU gene. To focus on the studied isolates in this work, it might be more facilitated to the readers by put some remark on the those isolates.  

The authors presented the phylogenetic trees of each gene and summarized the result of each gene in the supplementary table. 

Reviewer 2 Report

The subject is interesting and relevant. However, this is largely a repetition of existing data, confirming earlier conclusions. This is basically acceptable, as a review. However, the style of presentation is very poor. The trees (just neighbor joining, no bootstrap) are very unclear and superfluous. It would be much better if the authors would make a single table with the relevant data, i.e. the real results of strains having multilocus sequence data. This can be extracted from the supplementary tables. The trees are better deleted. Compared to the new data, the text is much too long. The authors correctly state that collection data should be updated, but giving these data in the text is user-unfriendly.